# Fumed Silica-Based Ultra-High-Purity Synthetic Quartz Powder via Sol–Gel Process for Advanced Semiconductor Process beyond Design Rule of 3 nm

**DOI:** 10.3390/nano13030390

**Published:** 2023-01-18

**Authors:** Ji-Ho Choi, Woo-Guk Lee, Tae-Hun Shim, Jea-Gun Park

**Affiliations:** 1Department of Electronic Engineering, Hanyang University, Seoul 04763, Republic of Korea; 2Department of Nanoscale Semiconductor Engineering, Hanyang University, Seoul 04763, Republic of Korea; 3Advanced Semiconductor Materials & Device Development Center, Hanyang University, Seoul 04763, Republic of Korea

**Keywords:** synthetic quartz powder, sol–gel process, ultra-high purity

## Abstract

Fumed silica-based ultra-high-purity synthetic quartz powder was developed via the sol–gel process to apply to quartz wares and quartz crucibles for use in advanced semiconductor processes. The process conditions of preparing potassium silicate solution, gelation, and cleaning were optimized, i.e., the relative ratio of fumed silica (10 wt%) to KOH (4 wt%) for potassium silicate solution, gelation time 3 h, and cleaning for 1 h with 5 wt% HCl solution. It was observed that the gelation time strongly affected the size distribution of the quartz powder; i.e., a longer gelation time led to a larger size (d50) of the synthesized quartz powder: 157 μm for 2 h and 331 μm for 5 h. In particular, it was found that the morphology of the as-synthesized quartz powder greatly depended on the pulverizing process; i.e., the shape of quartz powder was shown to be rod-shaped for the without-gel-pulverizing process and granular-shaped with the process. We expect that the fumed silica-based ultra-high-purity quartz powder with an impurity level of 74.1 ppb synthesized via the sol–gel process is applicable as a raw material for quartz wares and crucibles for advanced semiconductor processes beyond the design rule of 3 nm.

## 1. Introduction

For the past several decades, quartz wares for the semiconductor industry, which are made of quartz powder, have been widely used for consumable parts such as quartz crucibles, substrates, tubes, boats, rings, and baths in semiconductor equipment due to their high heat resistance (up to 1600 °C), high optical transmittance for ultraviolet and visible light wavelengths, and low thermal expanding coefficient (0.5 × 10^−6^/°C) [1,2,3,4,5]. When using such consumable quartz parts for the process equipment, they can easily generate particles and gases of metal impurities when they are exposed to high-temperature and -pressure processes [6] because their raw material is purified from natural rocks with an impurity level of up to 10 ppm in metal ions such as Fe, Al, Ni, and Zn. Recently, since the channel length of metal–oxide–semiconductor field-effect transistors (MOSFETs) has been scaled down to 3 nm to achieve faster speed, lower power consumption, and higher integration density [7], ultra-high-purity synthetic quartz powder with an impurity level of less than 100 parts per billion (ppb) in metal ions is essential so that the fabricating devices are not contaminated by metallic impurities generated from such quartz parts during the fabrication process [8,9]. In addition, for quartz crucibles used in Czochralski technology, a 3 nm thick inner layer of quartz crucible is fabricated by fusing and coating synthetic quartz powder instantaneously on the inner surface of a quartz crucible formed of natural quartz through the arc plasma process. Since the inner layer is in direct contact with molten poly-Si at a high temperature of ~1500 °C during the growth process, the purity level of the synthetic quartz powder is required to be more than 99.9999% (6 nines, 6 N) [10]. In particular, the inner layer of the crucible coated with the synthetic quartz powder should have a lower level of metallic impurity than 100 ppb because metallic impurities are a source of bubbles and defects in the inner layer and can easily affect the quality of the grown ingot [11]. To date, to synthesize quartz powder used in the semiconductor industry, the alkoxide silica-based sol–gel process was developed [12], where the silica sol–gel process through hydrolysis and condensation involves chemical reactions of tetraethyl orthosilicate (TEOS) or tetramethyl orthosilicate (TMOS) with ethanol and NH_4_OH [13,14,15]. However, this synthesis process has drawbacks in terms of maintaining consistent quality and competitive prices of quartz powder due to the process steps being complicated and the high material cost [16]. In addition, the synthesized quartz powder derived from TEOS contains residual carbon impurities that generate bubbles up to 100 μm in size when the quartz glass and quartz wares are manufactured. Since organic material (i.e., ethanol) is used to induce the condensation and hydrolysis of TEOS or TMOS, the bubbles can be formed by CO_2_ and CO gases due to residual carbon in the quartz powder during the melting process at a high temperature. This adversely affects the transmittance and the heat resistance properties of quartz parts and acts as a source of defects in quartz crucibles, resulting in defects in Si ingots [17]. Therefore, there are studies to minimize the carbon content in silica by controlling the heat-treatment process conditions [18,19], but a small amount of residual carbon inevitably remains due to the use of organic materials.

In this study, therefore, we develop a novel fumed silica-based sol–gel process for synthesizing ultra-high-purity quartz powder to achieve the requirements of semiconductor technology with a design rule of 3 nm, i.e., a metal impurity level of less than 100 ppb and median size distribution (d50) of the powder of about 150–200 μm. In particular, the optimal process conditions for the preparation of potassium silicate solution, gelation time, and cleaning process of the as-synthesized quartz powder were determined by analyzing the characteristics of the as-synthesized quartz powder. In addition, we investigated how the characteristics of potassium silicate solution are affected by the relative concentration ratio of fumed silica and KOH. Furthermore, we investigated the effect of gelation time and gel-pulverizing process on the morphology and characteristics of as-synthesized quartz powder by studying the mechanism of the gelation process in the formation of quartz powder. Moreover, the effect of hydrochloric acid (HCl) cleaning on reducing the metal impurity of the synthesized quartz powder was investigated.

## 2. Materials and Methods

### 2.1. Synthesis Process of Quartz Powder

The novel process flow for synthesizing ultra-high-purity quartz powder based on fumed silica is presented in Figure 1. The process involves the steps of preparing the potassium silicate solution, ion exchange, gelation, freezing, thawing, drying, cleaning, drying, and sintering. First, 500 g of potassium silicate solution was prepared by mixing with 6 wt% fumed silica, 2.4 wt% KOH, and 91.6 wt% deionized water. The solution was poured into polypropylene (PP) bottles and placed on hotplates at 80 °C for 24 h under 300 rpm of mechanical stirring via a magnetic stirrer. The mixing ratio among fumed silica, KOH, and deionized water was determined by the experiment for the dependency of dissociation of fumed silica on the weight percent of KOH and fumed silica. For the ion-exchange process, 300 g of strongly acidic cation exchange resin was put into a PET cartridge, and 500 g of the potassium silicate solution was ion exchanged through the cartridge at a flow rate of 188 mL/min. The potassium silicate solution was changed to SiO_2_ sol state via the ion-exchange process and then gel state through heating on a hotplate at 60 °C for 3 h. To investigate the effect of the pulverizing process on the morphology and crystallinity of the quartz powder, two gel samples in PP bottles were prepared. One of the samples was preceded by pulverizing via the bottle-shaking method, and the other was not. Then, two gel samples were frozen at −25 °C for 16 h, followed by thawing at 80 °C for 4.5 h, and were turned into quartz powder form through the decantation and drying process. In addition, the experiments on the dependency of forming as-synthesized powders were carried out after the freezing and thawing process on gelation time ranging from 0 to 5 h. Furthermore, to investigate the effect of the HCl cleaning process on reducing the metal impurity of the synthesized quartz powder, 10 g of each quartz powder sample was taken and cleaned via stirring at 300 rpm at 60 °C for 1 h in 500 g of 5 wt% HCl solution. After the cleaning process, the synthesis process of quartz powder was completed through an additional sintering process.

### 2.2. Characterization

The transmittance spectra of potassium silicate solutions at wavelengths ranging from 300 to 1000 nm and at wavenumbers ranging from 500 to 2000 cm^−1^ were measured via UV–visible spectroscopy (Cary 5000, Agilent Technologies Co., Inc., Santa Clara, CA, USA) and Fourier-transform infrared spectroscopy (FT-IR, Nicolet iS50, Thermo Scientific Co., Inc., Waltham, MA, USA) with attenuated total reflection (ATR) mode, respectively. After the gelation process, the morphology and crystallinity of the two groups of quartz powder samples were analyzed using scanning electron microscopy (SEM) and X-ray diffraction (XRD, D8 ADVANCE, Bruker Co., Inc., Billerica, MA, USA), respectively. The absorbance spectra of the hydrogels and as-synthesized powder samples were analyzed at wavelengths ranging from 500 to 2000 cm^−1^ via FT−IR spectroscopy with ATR mode. In addition, the size distribution of the quartz powder samples with various gelation times was characterized using a particle size distribution analyzer (LA-960, HORIBA Scientific Co., Inc., Kyoto, Japan). The impurity level of each element of quartz powder was analyzed by inductively coupled plasma mass spectroscopy (ICP-MS, Agilent 8800, Agilent Technologies Co., Inc., Santa Clara, CA, USA), where the detection limit of ICP-MS was >ppt.

## 3. Results and Discussion

### 3.1. Chemical Properties of Potassium Silicate Solutions Depending on KOH and SiO_2_ Concentration

The water glass typically used in industry is synthesized using sodium silicate and potassium silicate. In our experiment, as the selective adsorption of the strongly acidic cation exchange resin is greater when the valence of the ion is larger, the water glass was synthesized via a KOH solution rather than a NaOH solution to use a potassium ion with a relatively large valence in terms of efficiency [20]. When synthesizing fumed silica-based quartz powder, we prepared a potassium silicate solution mixed with fumed silica, KOH, and deionized water (DIW), where the commercial fumed silicas included the metallic concentration of ~1.292 ppm, as shown in Appendix A. The chemical reaction between KOH and fumed silica, which synthesizes a potassium silicate solution, is given in Equation (1). In addition, the chemical reaction to remove potassium ions from the synthesized potassium silicate solution during an ion exchange process with a strongly acidic cation-exchange resin is expressed in Equations (2) and (3).
2KOH + SiO_2_ + H_2_O → K_2_SiO_3_ + 2H_2_O(1)
4R−SO_3_H + 2K_2_SiO_3_ + nH_2_O → 4R−SO_3_^−^ + 4H^+^ + 4K^+^ + 2SiO_2_ + 2O_2_^−^ + nH_2_O(2)
4R−SO_3_^−^ + 4H^+^ + 4K^+^ + 2SiO_2_ + 2O_2_^−^ + nH_2_O → 4R−SO_3_K + 2SiO_2_ + nH_2_O(3)

Figure 2 shows the dependency of the dissociation of fumed silica on the weight percent of KOH ranging from 1 to 10 wt% at a fixed fumed silica concentration of 10 wt%. As is shown in Figure 2a, the samples of as-synthesized solutions look opaque to the naked eye because SiO_2_ is not completely dissociated within the KOH solution, which means that the chemical reaction between SiO_2_ and KOH is not finished. On the other hand, the samples with a KOH concentration more than 4 wt% became transparent after 24 h. These transparencies of the water glass samples after 24 h were well-correlated with the transmittance data characterized by UV–visible spectroscopy, as is shown in Figure 2b. This indicates that the transmittance of visible light for the samples of KOH concentrations more than 4 wt% were more than 99.9% and decreased gradually when the KOH concentration was less than 3 wt%.

To understand the change in the components of the sample solutions, the differences in the transmittance spectra for potassium silicate solutions at different KOH concentrations were observed at wavenumbers 500–2000 cm^−1^ using FT-IR, as is shown in Figure 2c. The two main peaks for the fumed silica (SiO_2_) and the water glass were observed at the 1100 and 1030 cm^−1^ bands, respectively. The peak at 1100 cm^−1^ corresponds to the Si−O−Si stretching vibration, which mostly appears in SiO_2_, and the peak at 1030 cm^−1^ corresponds to the Si−O (K) vibration, which mostly appears in water glass [21]. The intensity of the peak at 1100 cm^−1^ decreased when increasing the concentration of KOH in the water glass and disappeared at KOH concentrations higher than 4 wt%. Meanwhile, the intensity of the peak at 1030 cm^−1^ increased gradually with the KOH concentration in the water glass. Interestingly, the intensities of both peaks when the KOH concentration was 3 wt% were almost the same. In other words, the intensity of the peak at 1100 cm^−1^ is dominant at KOH concentrations lower than 3 wt%, while the intensity of the peak at 1030 cm^−1^ is dominant at KOH concentrations greater than 3 wt%. This tendency of the intensity of the peak at different KOH concentrations is obvious when observing the difference in the intensity of the Si−O−Si peak (1100 cm^−1^) and Si−O (K) peak (1030 cm^−1^), which are shown in Figure 2d. As the concentration of KOH in the water glass increased, the difference gradually decreased; in other words, the dominant peak at higher KOH concentrations than 3 wt% was switched from the Si−O−Si stretch peak to the Si−O (K) stretch peak. This was probably due to the reaction of the fumed silica (SiO_2_) with KOH to form K_2_SiO_3_, and higher concentrations of KOH led to a greater number of K_2_SiO_3_ bonds, as can be seen in Equation (1).

### 3.2. Effect of the Gel-Pulverizing Process on the Morphology of the As-Synthesized Quartz Powder

Figure 3 shows the effect of the gel-pulverizing process on the morphology of the as- synthesized quartz powders. Three hours after the ion-exchange process, SiO_2_ sol was changed into a single lump of gel, as is shown in Figure 3a. When subjected to the gel-pulverizing process, the gel lump is separated into many gel lumps of several mm (see Figure 3b). In addition, the morphology of the quartz powder synthesized via the gel-pulverizing process was very different from that produced without the gel-pulverizing process, as is shown in the SEM images in Figure 3c,d. That is, the morphology of the quartz powder synthesized from the gel without the pulverizing process was rod-shaped, and their widths and lengths were a few hundred micrometers and a few centimeters, respectively (see Figure 3c). On the other hand, the powder synthesized with the gel = pulverizing process was granular-shaped, and their widths and lengths ranged from 50 to 400 μm (see Figure 3d). Interestingly, however, no difference in the crystallinities of two groups of quartz powders were observed, as they showed the exact same peaks in the rocking curve of the XRD pattern as normal SiO_2_ (see Appendix A) [22,23].

The difference in the morphology of the quartz powder induced by the gel-pulverizing process can be explained by the modeled mechanism illustrated in Figure 4. After they ion-exchange process, the water glass, i.e., SiO_2_ sol, formed a three-dimensional polymer form as a result of the covalent bonding of SiO_2_ particles in the sol over 3 h and formed a hydrogel in which H_2_O molecules fill the extra space. In the case of the gel sample without the pulverizing process, H_2_O molecules present as liquid were partially solidified in the freezing process and formed a crystalline structure that resulted in SiO_2_ polymer chains in the gel being pushed out and cross-linked before being further polymerized via agglomeration [24,25]. In this process, since the gel in the bottle existed as a lump with a size of several centimeters or tens of centimeters, SiO_2_ polymer can grow in one direction and be synthesized to a rod-shaped morphology with a width of several millimeters and a length of several centimeters or more.

On the other hand, when the pulverizing process was performed, the SiO_2_ components were physically limited from growing in one direction when agglomeration occurred in the freezing process because the size of the gel mass reached up to 3 to 4 mm, which resulted in granular-shaped as-synthesized quartz powder being formed rather than rod-shaped quartz powder. That is, the difference in morphology shown in the SEM images of Figure 3 was due to the growth in one direction being limited by the pulverizing process. The reason why no differences in the crystalline of the quartz powders with and without the pulverizing process were observed is that the difference in morphology depending on whether the gel-pulverizing process was carried out or not was simply determined by a physical size limitation in the freezing and thawing processes.

### 3.3. Dependencies of Quartz Powder Formation on Gelation Time

The effect of the gelation time in the chemical reaction to change the phase from sol to gel on the gel and quartz powder states was investigated by observing the absorbance spectra of the gels and powders at wavenumbers of 500–4000 cm^−1^, as is shown in Figure 5. In the case of the gel state, four peaks appeared in the absorbance spectrum, which are shown in Figure 5a, i.e., a broad OH stretching peak at 3000–3700 cm^−1^ derived from the OH group, an OH bending peak at ~1640 cm^−1^, an asymmetric Si−O−Si stretching (V_as_Si−O−Si) peak at ~1075 cm^−1^, and an asymmetric O−Si−O bending (V_as_O−Si−O) peak at ~520 cm^−1^ [26,27,28]. Among the four peaks, the intensity of the peak at ~1075 cm^−1^, V_as_Si−O−Si, gradually increased with the gelation time, as is shown in Figure 5b. The increase in the intensity of the V_as_Si−O−Si peak was because the Brownian motion of the sol molecules became active through high energy at a temperature of 60 °C, and polymerization occurred through the aggregation of SiO_2_ particles in an unstable state over time during the gelation process after ion exchange. In other words, as is shown in Appendix A, SiO_2_ monomers of tetrahedral structure are gradually polymerized into Q1, Q2, and Qn in the gelation process to form a polymer chain, and as they are polymerized, fluidity is lost, and a gel is formed instead of a sol [29]. Therefore, since SiO_2_ polymer chains are formed as the gelation process time increases, the total number of Si−O−Si bonds in the sample increased, which resulted in a gradual increase in the intensity of V_as_Si−O−Si in the absorbance spectrum.

The quartz powder samples were synthesized through the freezing, thawing, and drying of the gel samples used in Figure 5a. Six main peaks appeared in the absorbance spectrum of the quartz powder samples, as shown in Figure 5c, i.e., a broad OH stretching peak at ~3300 cm^−1^ derived from the OH group, an OH bending peak at ~1640 cm^−1^, an asymmetric Si−O−Si vibration (V_as_Si−O−Si) peak at ~1050 cm^−1^, a Si−O_4_ vibration peak (VSi−O_4_) at ~955 cm^−1^, a symmetric Si−O−Si vibration (V_s_Si−O−Si) at ~795 cm^−1^, and an asymmetric O−Si−O bending (V_as_O−Si−O) peak at ~550 cm^−1^ [30]. The change in the intensity of the six main peaks at different gelation times is shown in Figure 5d. In the cases of samples synthesized with a hydrogel form, i.e., a gelation time of 0 or 1 h, a higher intensity of the OH stretching and OH bending peaks was observed than in the samples processed with a gelation time of 2, 3, 4, and 5 h, which were synthesized in granular-shaped powder form, as shown in Appendix A. This was caused by the remaining moisture caused by the insufficient separation of the SiO_2_ component and H_2_O during the freezing and thawing processes due to the insufficient gelation time of the SiO_2_ sol after ion exchange. The low intensities of the OH stretching and OH bending peaks observed at the samples processed at a gelation time of 2, 3, 4, and 5 h were probably due to OH− binding to the powder’s surface. Interestingly, the intensities of the V_as_Si−O−Si peak at ~1050 cm^−1^, the VSi−O_4_ peak at ~955 cm^−1^, the V_s_Si−O−Si peak at ~795 cm^−1^, and the V_as_O−Si−O bending peak at ~550 cm^−1^ gradually decreased as the gelation time increased. This tendency for the intensity of the four peaks to decrease as the gelation time increased can be explained by the fact that the SiO_2_ chain in the powder was strengthened, and more moisture was separated from the gel via syneresis as the gelation time increased [31], which resulted in a decrease in the intensity of the peaks related with the Si−O−Si and O−Si−O bonds due to a decrease in the number of O molecules.

The dependency of the size distribution of the as-synthesized quartz powder on gelation time was investigated, and the size distributions of the powder samples processed with different gelation times are presented in Figure 6. As the gelation time increased from 2 to 5 h, the size (d50) of the quartz powder samples gradually increased from 159 to 364 μm. The values of the median, mean, range, and standard deviation at different gelation times are summarized in detail in Table 1. This increase in the size of powders at different gelation times was probably because the sol attained higher-strength SiO_2_ bonds during gelation after ion exchange and thus formed larger gel lumps in the gel-pulverizing process. In addition, more crosslinks with the SiO_2_ component were formed in the freezing process because more polymerization was generated in the gel state.

### 3.4. Effect of Hydrochloric Acid Solution Cleaning on the Reduction of Metallic Impurities for Quartz Powder

To satisfy the level of impurity required for semiconductor processes, the effect of HCl cleaning frequency on the metallic ion impurity concentration of the as-synthesized powders was investigated (see Figure 7). The total impurity level in the as-synthesized quartz powder samples was 1874 ppb; the main metal elements of impurity were Na, Al, K, and Fe; and their levels were 30.7, 286.9, 206.4, and 650.3 ppb, respectively. After cleaning once with 5 wt% HCl solution for 1 h, the total impurity level was significantly reduced to a level of 74.1 ppb. In particular, the impurity levels of Na and K ions were completely removed to a level of less than 10 ppb, and impurity levels of Al and Fe ions decreased to 53.5 and 14.5 ppb, respectively. In general, HCl is known to be the most effective reagent for metal dissolution compared to other acid chemicals, such as H_2_SO_4_ and HNO_3_, because HCl is capable of a high degree of metal dissolution due to the high affinity of Cl^−^ for metal elements. This impurity reduction caused by HCl solution is generated through the process of chemically reacting and dissolving with metal impurities, as is shown in Equations (4)–(6) [32,33,34,35,36].
FeOOH + 4HCl → FeCl_2_ + 2H_2_O + Cl_2_(4)
Fe_2_O_3(s)_ + 6H^+^ + Cl^−^ → 2Fe^3+^ + 6Cl^−^ + 3H_2_O_(l)_(5)
Al(OH)_3(s)_ + 3HCl_(aq)_ → AlCl_3(aq)_ + 3H_2_O(6)

Since each impurity element in the quartz powder has a different bonding state, the effect of the HCl cleaning process was varied depending on the impurity element. In general, Na^+^ and K^+^ impurities in quartz are present at interstitial channel positions, and Al is present as a substitutional structure of Si^4+^. In the case of Fe, Fe^2+^ exists as an interstitial and Fe^3+^ as a substitutional structure [37,38,39]. It is noted that the impurities (i.e., Na^+^, K^+^, and Fe^2+^) that exist in interstitial positions react relatively easily with HCl solution during the HCl cleaning process and are effectively removed. On the other hand, impurities (i.e., Fe^3+^ and Al^3+^) that exist as substitutional structures are relatively difficult to remove through the HCl cleaning process because their atoms are substituted Si^4+^ and bonded, and all covalent bonds with adjacent oxygen atoms must be cut off to be removed through the HCl cleaning process. For these reasons, when cleaning once with HCl, the Na and K impurities were completely removed except for a small amount of less than 10 ppb, whereas Fe and Al residual impurities of 14.5 and 53.5 ppm remained, respectively. In addition, there was no significant difference in the impurity reduction effect with different HCl cleaning frequencies. This is because Al and Fe elements substituted deep inside the SiO_2_ particle in the process of forming quartz powder through polymerization and agglomeration are unlikely to break their bonds, so it is unlikely that they will react with HCl and elute even if HCl cleaning is carried out three times. As a result, a synthesized quartz powder with total metallic impurities of 74.1 ppb was achieved with HCl cleaning for 1 h, indicating that this level of purity can satisfy the requirements of advanced semiconductor processes beyond design rule 3 nm. In particular, a synthesized quartz powder in this experiment showed far superior purity than other amorphous silica particles, as shown in Appendix A [40,41,42,43,44].

## 4. Conclusions

As the channel length of MOSFETs for logic devices has been scaled down to 3 nm, the total impurity level of quartz parts such as quartz tubes, boats, rings, and crucibles is essential to control for less than 100 ppb to avoid wafer contamination during the high-temperature processes. As a raw material of such quartz parts, ultra-high-purity synthetic quartz powder was developed via fumed silica-based sol–gel process while optimizing key process conditions. The relative ratio of fumed silica to KOH was tested at 10 to 4 wt% by observing the difference in the transmittance spectra at wavenumbers ranging from 500 to 2000 cm^−1^. In particular, the pulverizing process made the synthesized quartz powders granular-shaped, which is a form commonly used in industry. In addition, it was observed that the gelation time greatly influenced the formation of both the gel and powder states. That is, as the gelation time to form the gel increased, the intensity in the V_as_Si−O−Si peak of gels increased gradually because SiO_2_ polymer chains were formed over time during the gelation process, and as a result, the total number of Si−O−Si bonds in the sample increased. Furthermore, it was observed that the four peaks (i.e., V_as_Si−O−Si at ~1050 cm^−1^, VSi−O_4_ at ~955 cm^−1^, V_s_Si−O−Si at ~795 cm^−1^, and V_as_O−Si−O bending at ~550 cm^−1^) of the synthesized quartz powder samples gradually decreased as the gelation time increased. This is because SiO_2_ chains in the powder were strengthened, and more moisture was separated from the gel during the gelation process, which resulted in a decrease in the intensity of the peaks related to Si−O−Si and O−Si−O bonds due to a decrease in the number of O molecules. The gelation process time affected the size (d50) of the quartz powder samples. Quartz powders with total metal impurity levels of less than 74.1 ppb were synthesized using only HCl cleaning for 1 h.

Ultra-high-purity quartz powders having less than the heavy metal concentration of 100 ppb can be essentially employed to coat an inner layer of a high-purity quartz crucible for growing high-purity silicon ingot via the Czochralski method. In addition, ultra-high-purity quartz powder can be fundamentally utilized as a raw material producing the high-purity quartz parts used in semiconductor equipment such as dry etchers, chemical vapor deposition reactors, and oxidation furnaces. Note that the presence of high-purity Si ingots and quartz parts of semiconductor process equipment directly and remarkably affects the yield of the devices fabricated with Si ingots and quartz powder; generally, the requirement level of remaining heavy-metallic impurity concentration in the semiconductor devices has been lower and lower when the design rule of the semiconductor devices has been scaled down.

## Figures and Tables

**Figure 1 nanomaterials-13-00390-f001:**
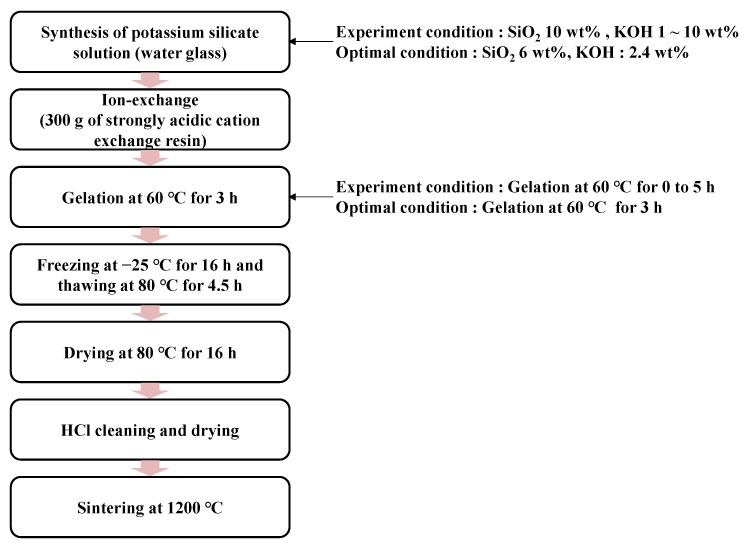
Synthesis process flow of the fumed silica-based ultra-high-purity quartz powder.

**Figure 2 nanomaterials-13-00390-f002:**
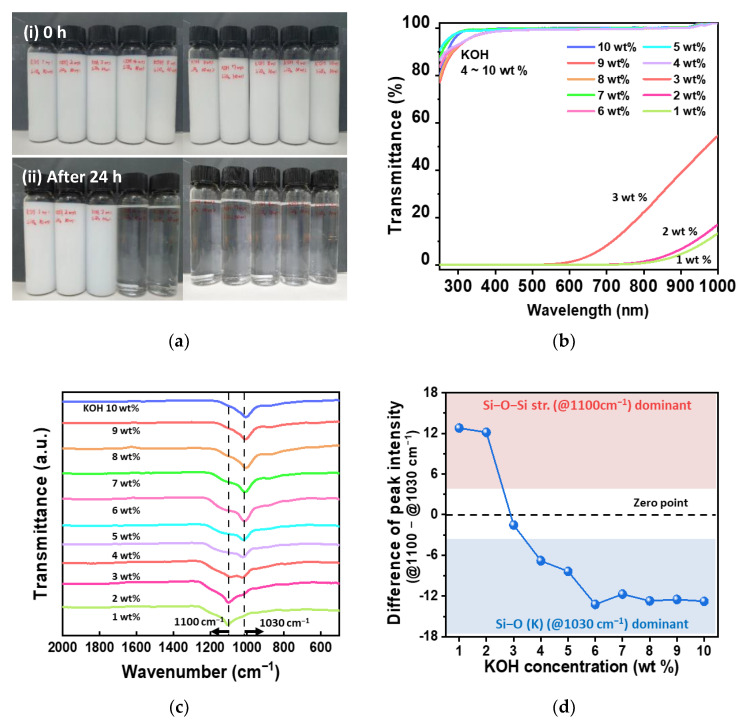
Dependency of dissociation of fumed silica on KOH weight ratio. (**a**) Photographs of potassium silicate solutions (i) before and (ii) after 24 h. The KOH concentrations of the solutions are 1 to 10 wt% in order from the left side of the photographs. (**b**) Transmittance spectra at wavelengths of 300−1000 nm measured by UV–visible spectroscopy. (**c**) Transmittance spectra at wavenumbers of 500−2000 cm^−1^ measured by FT−IR. (**d**) Difference of peak intensity at vibration of Si−O−Si peak (1100 cm^−1^) and Si−O (K) (1030 cm^−1^) peak.

**Figure 3 nanomaterials-13-00390-f003:**
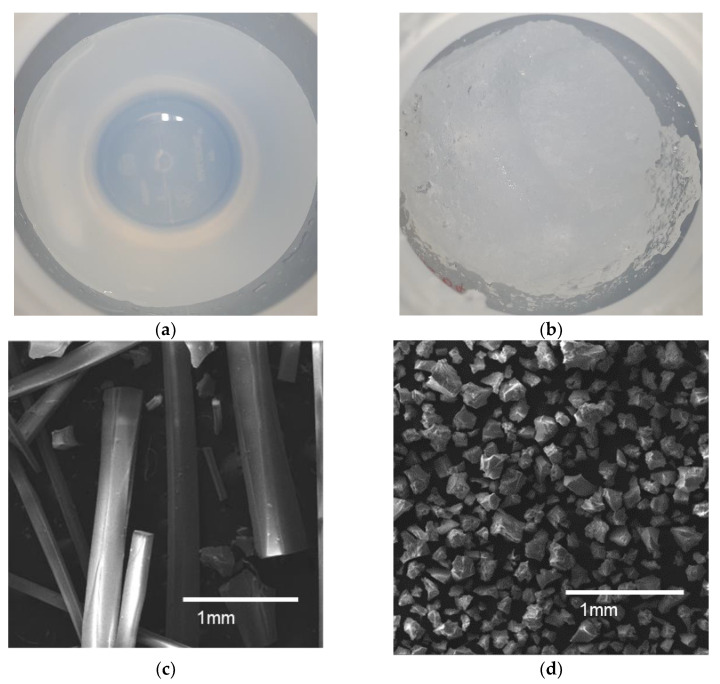
Effect of gel-pulverizing process on morphology of as-synthesized quartz powder. (**a,b**) Photographs of hydrogels with and without pulverizing process, respectively. (**c,d**) SEM images of quartz powder samples synthesized with and without pulverizing process, respectively.

**Figure 4 nanomaterials-13-00390-f004:**
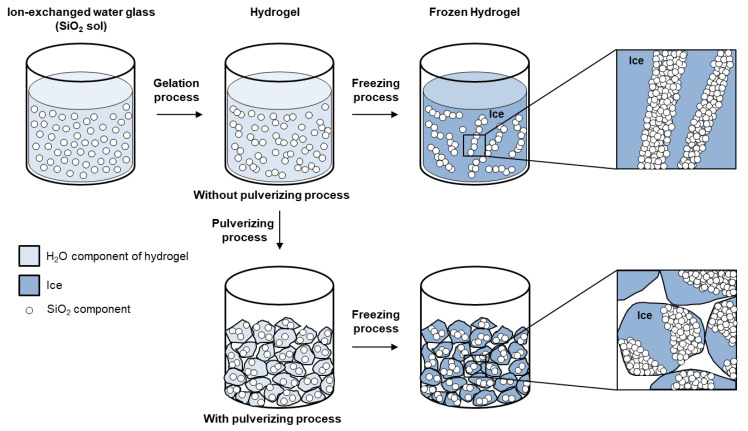
Schematic model of forming morphology of as-synthesized quartz powder.

**Figure 5 nanomaterials-13-00390-f005:**
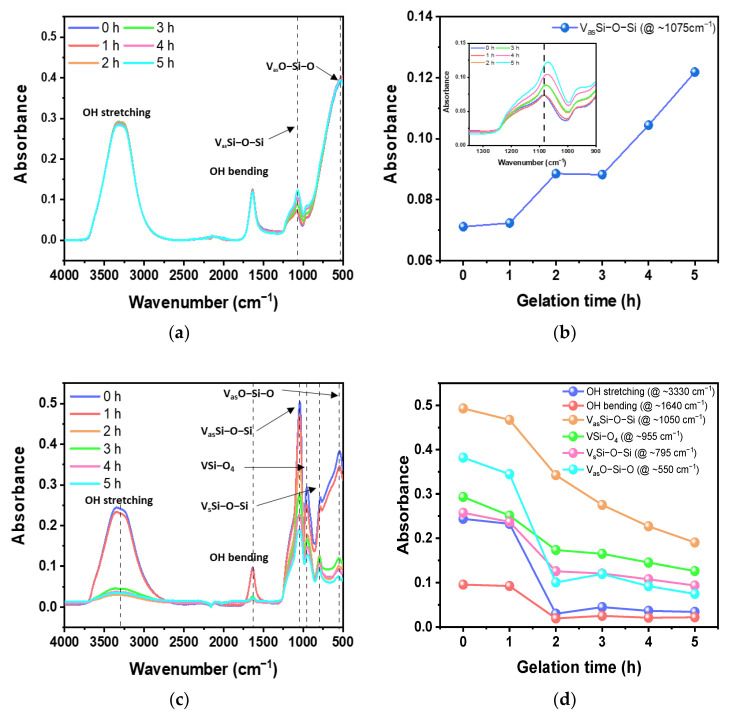
Dependency of absorbance spectrum of gel and powder on gelation time from 0 to 5 h, as analyzed by FT−IR. (**a**) Absorbance spectra of gel at wavenumbers of 500−4000 cm^−1^. (**b**) Peak intensity of gel at V_as_Si−O−Si (~1075 cm^−1^). The inset is a magnification of the absorbance spectra at wavenumbers of 1350−900 cm^−1^ (**c**) Absorbance spectra of powder at wavenumbers of 500−4000 cm^−1^. (**d**) Peak intensity of powder at the OH stretching (~3330 cm^−1^), OH bending (~1640 cm^−1^), V_as_Si−O−Si (~1100 cm^−1^), VSi−O_4_ (~950 cm^−1^), V_s_Si−O−Si (~790 cm^−1^), and V_as_O−Si−O (~550 cm^−1^) peaks.

**Figure 6 nanomaterials-13-00390-f006:**
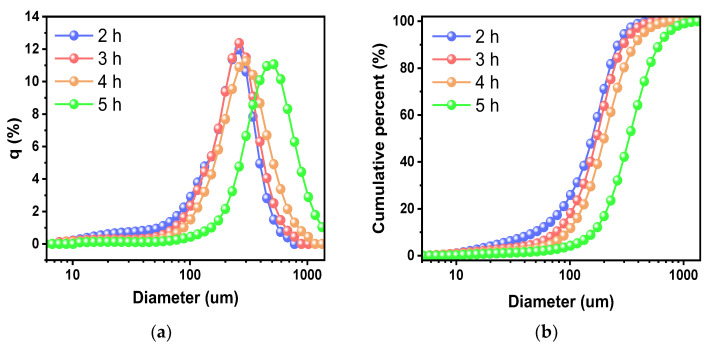
Dependency of size distribution of as-synthesized powder on gelation time ranging from 2 to 5 h. (**a**) Particle size distributions. (**b**) Cumulative particle size distributions.

**Figure 7 nanomaterials-13-00390-f007:**
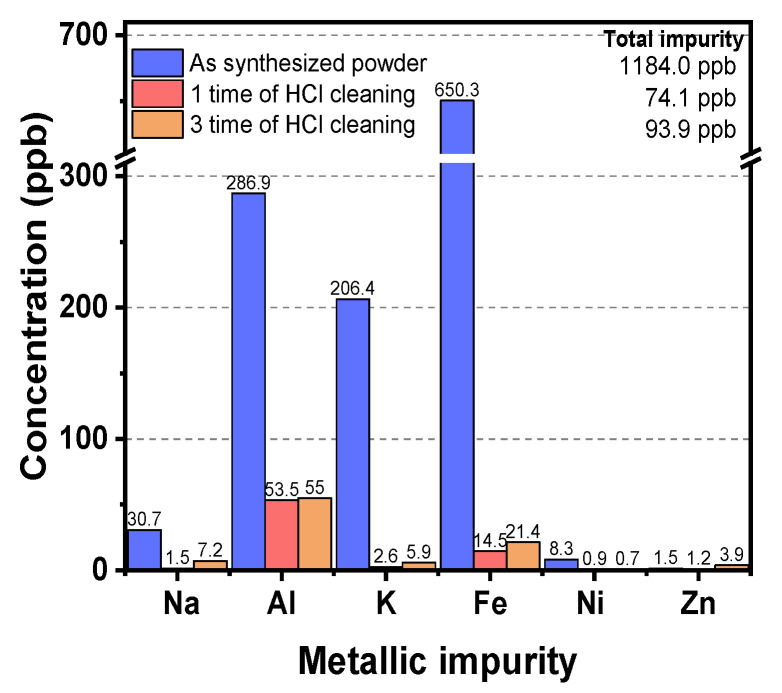
Effect of the HCl cleaning process on reduction of the metal impurity concentration of as-synthesized powder.

**Table 1 nanomaterials-13-00390-t001:** Dependency of size distribution characteristics on gelation time.

	2 h	3 h	4 h	5 h
Median size (μm)	157	172	200	331
Mean size (μm)	159	181	220	364
Variance (μm^2^)	7200	8320	13,770	36,907
St. Dev. (μm)	85	187	117	192

## Data Availability

Not applicable.

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
