# Peer review of "Fumed Silica-Based Ultra-High-Purity Synthetic Quartz Powder via Sol–Gel Process for Advanced Semiconductor Process beyond Design Rule of 3 nm"

_nanomaterials, 2023, doi:10.3390/nano13030390_

Round 1

Reviewer 1 Report

The article describes a way to prepare a 150 micron sized particle of silica suitable for chemomechanical polishing (CMP).  The method involves the incorporation of fumed silica in a potassium silicate (soluble silicate) solution.  The potassium is removed by an ion exchange.  The claim is that the powders are purer than others derived from alkoxy sources.  The paper reads more like a patent application than a research paper.

There are 3 concerns that that the authors should address:

What is the source of the fumed silica?  Is it a commercial fumed silica?  If so, what are the impurities that the supplier indicates for this powder?  Did the authors try other sources of fumed silica?  Is the behavior generally observed with commercial fumed silicas or is it particular to this one type of fumed silica?

What are the detection limits?  When the impurity levels are determined for Na, Al, K and Fe, what are the limits given by the instrument maker?  Can these values really be accurate to 4 figures in ppb? 

How does this process relate to the “Shoup Process”?  Robert Shoup from Corning, Inc., developed a soluble silicate method in the 1980’s.  For example, DURABLE GLASS BY RECONSTITUTION OF HYDRATABLE SODIUM-SILICATE GLASSES, BARTHOLOMEW, R, HAYNES, W, SHOUP, R, ACS Symposium Series, Volume194, pp. 277-289, 1982, and US Patent 4,059,658 cover a soluble silicate approach to quartz powder.  Please discuss.

If the authors can explain these points, then the paper is publishable.

Author Response

Response 1: Thank you for your comment. It is a commercial fumed silica specified by including SiO2 >99.8 wt%, Al2O3 <0.05 wt%, and Fe2O3 <0.003 wt%. Thus, we analyzed the impurities of the commercialized fumed silicas used in our experiments, as shown in Supplementary Fig. 1(a). There total impurity concentration was 1.292 ppm. In addition, we produced the synthesized quartz powder using high-purity colloidal silicas, as shown in Supplementary Fig. 1(b). Although the colloidal silicas include much lower metallic impurities than the fumed silica, our study used the fumed silicas since the cost of the fumed silica was ~3 times lower than that of the colloidal silicas. Thus, we would like to modify our manuscript as follows;

Response 2: The detection limit of the ICP-MS (Agilent 8800, Agilent Technologies Co., Inc., Santa Clara, USA) used in our experiment was >1 ppt. Thus, we would like to modify our manuscript as follow.

Response 3: There are two significant differences between the Shoup process and the synthesis process in this study. First, the method of condensing an alkali silicate solution (or glass in the Shoup process) to form a bulk body in a hydrogel state is different. In our work, the SiO2 hydrogel bulk body was obtained by completely removing alkali metals via ion exchange of potassium silicate solution and the gelation process. Therefore, the total impurity level of the synthesized quartz powder is about 74.1 ppb, and Na+ and K+ contents are very low at 1.5 and 2.6 ppb, respectively. On the other hand, in the Shoup process, sodium silicate glass is hydrated, or sodium silicate solution is dehydrated to form a hydro-silicate body, which is de-alkalized through a leach solution. Since the bulk body is already formed via the hydrating process, it is difficult to remove alkali metal in the bulk body completely. Therefore, as a result, the alkali metal impurity in the synthesized glass appears at the level of 100 ppm.

Second, the process of converting the bulk of the hydrogel state into a glass form is different, resulting in a difference in the form of the final compound (powder form or glass form). In this study, SiO2 dry gel is obtained in powder form by separating the SiO2 component and moisture via the freezing and thawing processes of the hydrogel, and quartz powder is synthesized through high-temperature heat treatment at 1200℃(sintering). On the other hand, the Shoup process dries the porous body formed by de-alkalizing the hydro-silicate body and forms transparent fused glass through high-temperature heat treatment (consolidation).

In addition, the final product of the Shoup process and this study have different applications. Quartz powder synthesized by our method is used as a material for quartz parts (i.e., quartz crucibles, substrates, boats, tubes, rings, baths) to be used in the semiconductor industry is synthesized, and, therefore, to prevent contamination of semiconductor devices, ultra-high purity of less than 1 ppm of total metallic impurities is required. On the other hand, the Shoup process synthesizes transparent dense high silica glass rather than raw materials for various types of quartz parts for semiconductor processing.

Reviewer 2 Report

Comments on the paper

Concerning Introduction and carbon content in sol-gel materials derived from TEOS, I just want to mention for the authors that according to some SIMS investigations of sol-gel derived films, the carbon content depends on the annealing conditions, and additional heat treatment at 600 °C results in depletion of spin-on silica sol-gel derived films with carbon [Gaponenko N. V., Gnaser H., Becker P., Grozhik V. A. Carbon depth distribution in spin-on silicon dioxide films // Thin Solid Films. – 1995. – Vol. 261, N  1–2. – P. 186–191 ].

Erratum in Introduction: “cost. [16]”.

I recomend adding a brief comment to the caption of Fig. 2a stating that the concentration of KOH increases (decreases) from left to right or vice versa.

The  data presented in table Table 1 (Dependency of size distribution characteristics on gelation time) should be rounded off like in the text above the table like 159 and 364 instead of 159.49 and 364.06 microns while standard deviation is above tens of microns.

Author Response

Response 1: Thank you for your comment. We added a comment and reference in the introduction that there are studies to minimize the carbon content in TEOS-based silica materials by controlling the heat treatment process conditions. And also we explained the remaining limitation of TEOS-based synthesized silica. Thus, we would like to modify our manuscript as follow.

Response 2: We revised “cost. [16]” to “cost [16].”. Thank you.

Response 3: We inserted an additional comment to the caption of Fig. 2a as you mentioned.

Response 4: We revised all data presented in Table 1 by rounding them from one decimal place.

Reviewer 3 Report

This paper deals with the development of a process to obtain ultra-high-purity synthetic quartz powder for semiconductor applications. There are numerous characterizations, the work is very well-executed. The figures are well presented. The paper only needs minor revisions before publication, mainly for the presentation of the text. I found the text really dense and not so pleasant to ready. For me, the materials and methods must be separated in a synthesis process section and a characterization section. Do not hesitate to structure it in paragraphs to give a bit of breathing to the reader. For the results and discussion section, subsections with subtitles must be made with also different paragraphs to make the text more readable.

Author Response

Response 1: Thank you for your comment. We would like to modify the Materials and Methods as follow.

Response 2: For the results and discussion section, based on your comments, we have divided the content into 4 subsections as follows. In addition, we separated paragraphs to make the text more readable.

Reviewer 4 Report

1. The high-purity Spherical Silica Nanoparticles from Powder Quartz silica topic was widely investigated in the literature and the authors claimed that they “develop a novel fumed silica-based sol–gel process for synthesizing ultra-high-purity quartz powder” with several advantages. Are the authors aware about similar studies about the purity/impurity level of silica quartz powder prepared by sol-gel (or other method), for comparison?

2. In the conclusions is better to be extend the advance and the importance of the new knowledge acquired

Author Response

Response 1: Thank you for your comment. We found similar studies; thus, we would like to modify our manuscript as follow.

Response 2:  In the conclusions, we added some paragraphs to extend the advance and the importance of the new knowledge acquired as follows. 
